# Compensatory Transcriptional Response of *Fischerella thermalis* to Thermal Damage of the Photosynthetic Electron Transfer Chain

**DOI:** 10.3390/molecules27238515

**Published:** 2022-12-03

**Authors:** Pablo Vergara-Barros, Jaime Alcorta, Angélica Casanova-Katny, Dennis J. Nürnberg, Beatriz Díez

**Affiliations:** 1Department of Molecular Genetics and Microbiology, Biological Sciences Faculty, Pontifical Catholic University of Chile, Santiago 8331150, Chile; 2Millennium Institute Center for Genome Regulation (CGR), Santiago 8370186, Chile; 3Laboratory of Plant Ecophysiology, Faculty of Natural Resources, Campus Luis Rivas del Canto, Catholic University of Temuco, Temuco 4780000, Chile; 4Institute of Experimental Physics, Freie Universität Berlin, 14195 Berlin, Germany; 5Dahlem Centre of Plant Sciences, Freie Universität Berlin, 14195 Berlin, Germany; 6Center for Climate and Resilience Research (CR)2, Santiago 8370449, Chile

**Keywords:** thermophiles, cyanobacteria, *Fischerella thermalis*, photosynthesis, hot springs, photosystem II

## Abstract

Key organisms in the environment, such as oxygenic photosynthetic primary producers (photosynthetic eukaryotes and cyanobacteria), are responsible for fixing most of the carbon globally. However, they are affected by environmental conditions, such as temperature, which in turn affect their distribution. Globally, the cyanobacterium *Fischerella thermalis* is one of the main primary producers in terrestrial hot springs with thermal gradients up to 60 °C, but the mechanisms by which *F. thermalis* maintains its photosynthetic activity at these high temperatures are not known. In this study, we used molecular approaches and bioinformatics, in addition to photophysiological analyses, to determine the genetic activity associated with the energy metabolism of *F. thermalis* both in situ and in high-temperature (40 °C to 65 °C) cultures. Our results show that photosynthesis of *F. thermalis* decays with temperature, while increased transcriptional activity of genes encoding photosystem II reaction center proteins, such as PsbA (D1), could help overcome thermal damage at up to 60 °C. We observed that *F. thermalis* tends to lose copies of the standard G4 D1 isoform while maintaining the recently described D1^INT^ isoform, suggesting a preference for photoresistant isoforms in response to the thermal gradient. The transcriptional activity and metabolic characteristics of *F. thermalis*, as measured by metatranscriptomics, further suggest that carbon metabolism occurs in parallel with photosynthesis, thereby assisting in energy acquisition under high temperatures at which other photosynthetic organisms cannot survive. This study reveals that, to cope with the harsh conditions of hot springs, *F. thermalis* has several compensatory adaptations, and provides emerging evidence for mixotrophic metabolism as being potentially relevant to the thermotolerance of this species. Ultimately, this work increases our knowledge about thermal adaptation strategies of cyanobacteria.

## 1. Introduction

The global optimal temperature for oxygenic photosynthesis is estimated to be between ~10 °C and ~30 °C [1,2,3,4]. While oxygenic photosynthesis can occur at higher temperatures [5,6,7,8,9], it is known to be strongly affected by temperature, demonstrating a known upper limit of 75 °C [10], mostly owing to the sensitivity of its protein–pigment complexes [11,12,13,14]. Some cyanobacteria, such as *Fischerella thermalis*, have adapted to high temperature conditions [5,6,7,15] and can sustain the primary production requirements of microbial communities in these environments.

*Fischerella thermalis* (also known as *Mastigocladus laminosus*) was first described in a hot spring in the Czech Republic [16]. It is an oxygenic photosynthetic organism with high metabolic versatility that can grow autotrophically or mixotrophically [17,18] using different nitrogen sources, including N_2_ [8,18,19,20]. Environmental and laboratory data on *F. thermalis* have defined its optimal growth range between 45 °C and 55 °C, with an estimated photosynthetic limit near ~58 °C [8,15,19,20,21,22,23,24]. Moreover, sequences belonging to *F. thermalis* have been detected at temperatures as high as 66 °C [8,19], but its photosynthetic activity at and above this temperature has not yet been demonstrated, neither in the environment nor in the laboratory.

The thermotolerance of its protein complexes makes *F. thermalis* an interesting organism for structural studies of the photosynthetic apparatus. Components of particular interest include an extremely thermostable ferredoxin that serves as the electron acceptor for photosystem I (PSI) [25]; proteins of the light-harvesting phycobilisome complex [26,27,28,29]; phycoerythrocyanin [30] and allophycocyanin-linker proteins [31]; the cytochrome b_6_f complex [32,33]; and PSI [34,35]. 

Of the cyanobacterial photosynthesis components, the least thermoresistant is photosystem II (PSII) [14], a macromolecular complex composed of several protein subunits and cofactors, including the water-oxidizing complex (WOC). The WOC is mainly coordinated by the proteins D1 (PsbA) and D2 (PsbD), with D1 being directly involved in the photochemical reaction and requiring the highest turnover rates of the complex [36,37]. In cyanobacteria, there are several currently known distinct D1 isoforms that have been grouped as G0 to G4, of which G3 and G4 are the most common [38]. Recently, two new D1 forms belonging to G4 have been proposed: D1^INT^ and D1^FR^, which are present in most Nostocales cyanobacteria [39], including *F. thermalis*. In other cyanobacteria, such as *Thermosynechococcus elongatus* BP-1, it has been reported that the D1 isoforms are expressed differently in response to temperature [40]; however, their relationship to the thermal-adaptation process is still not well understood.

In cyanobacteria, photosynthesis is connected to aerobic respiration and carbon catabolism by participating in photorespiration and regulating photosynthesis through redox modulation in thylakoid membranes [41,42]. It has been suggested that these alternative pathways assist photosynthesis under stress conditions [43], but whether these energy pathways are involved in the thermal adaptation of *F. thermalis* is unknown. Understanding how these pathways are connected to *F. thermalis* photosynthesis could help to reveal how this organism adapted to high-temperature hot springs conditions.

Furthermore, the genus *Fischerella* harbors both thermophilic and mesophilic species [5]; thus, studying their evolutive history and physiological processes could help us better undestand the underlying adaptation of photosynthesis and its components to high temperatures. In this study, we analyzed how temperature affects the energy metabolism of *F. thermalis* using both environmental data and an isolated strain. The transcriptional activity in samples obtained from a thermal gradient of the Porcelana hot spring (Northern Patagonia, Chile) was evaluated, and gene coexpression in samples obtained from the El Tatio geothermal field (Atacama, Chile) was investigated. Additionally, the expression of key energy metabolism and photosynthetic efficiency genes was evaluated in temperature-treated axenic *F. thermalis* PCC 7521 cultures. We found that the transcriptional response of *F. thermalis* is consistent with a photosynthetic compensational response model, where increased transcription of energy metabolism genes helps to sustain *F. thermalis* photosynthesis at higher temperatures.

## 2. Results

### 2.1. Prevalence of the D1 Isoform in the Fischerella thermalis PSII Reaction Center

Most cyanobacteria have several copies and isoforms of the D1 protein [38]. These isoforms respond to environmental changes [36,40]; however, their relationship to high temperature adaptation is still poorly described. To determine the main D1 isoforms in the genus *Fischerella*, and particularly in *F. thermalis*, we reconstructed a phylogenetic tree of the *psb*A gene that encodes the D1 protein using 48 *Fischerella* genomes available in the NCBI RefSeq and GenBank databases [44,45] (Figure 1A). Previously identified D1 sequences from *Gloeobacter violaceus*, *Nostoc* sp., and *Chlorogloeopsis fritschii* [38,39,46] were used here as isoform outgroups, along with 30 *psb*D sequences. Our results show that the D1 isoforms G1–G4, including D1^INT^ and D1^FR^, are widespread within the genus *Fischerella*. Differences in amino acids have been previously reported between D1^INT^ and G4 associated with PSII internal electron acceptors [39]. Consequently, we found that the amino acid residues Trp126 and Trp260 (corresponding to Tyr126 and Phe260, respectively, in G4) [39] are conserved in all D1^INT^ sequences of the genus *Fischerella*. For D1^FR^, we found that amino acid residues related to chlorophyll *f* binding, such as Tyr120 and Thr155 [47,48], are also conserved. The gene content of *F. thermalis* is known to vary compared with closely related cyanobacteria, thereby generating phenotypic plasticity related to specific niches in a temperature gradient [44,45]. Thus, we analyzed the gene copy number of each isoform within the genus *Fischerella* (Figure 1B). Genomes belonging to the genus *Fischerella* were classified using the GTDB-tk software R202 and by phylogenomic reconstruction, being consistent with the 16S rRNA phylogeny (Appendix A). As a result, we found that the number of D1^INT^-*psb*A and D1^FR^-*psb*A remained stable within the genus, with ~1 copy per genome (Figure 1B and Appendix A). As most of the 48 *Fischerella* genomes were fragmented, we focused on the only *F. thermalis* closed genome (strain NIES-3754, GCF_001548455.1 [49]), which showed that *F. thermalis* has three G4-*psb*A copies (Appendix A). In contrast, the only complete genome from a mesophilic *Fischerella* (strain NIES-4106, GCF_002368315.1 [50]) has five copies of G4-*psb*A (Appendix A). Differences in the G4-*psb*A copy number between *F. thermalis* and the mesophilic *Fischerella* were also observed in the high-quality open genomes of the genus *Fischerella* (Appendix A), supporting that *F. thermalis* has fewer copies of G4-*psb*A than mesophilic *Fischerella*. Additionally, we compared the genomic context of G4-*psb*A and D1^INT^-*psb*A genes in the *F. thermalis* NIES-3754 and mesophilic *Fischerella* sp. NIES-4106 genomes (Figure 1C,D). The analysis showed that one of the G4-*psb*A copies in *F. thermalis* was close to D1^INT^-*psb*A, while another was close to genes encoding phycobilisome rod proteins (Figure 1C), suggesting some degree of G4 gene clustering in *F. thermalis*. Conversely, some G4-*psb*A of *Fischerella* sp. NIES-4106 were found near transposase genes (Figure 1D), possibly as part of mobile elements of the *Fischerella* sp. NIES-4106 genome. The genomic context of D1^INT^-*psb*A was analyzed in 45 genomes of the genus *Fischerella* where D1^INT^-*psb*A was detected (summarized in Appendix A). The results show that D1^INT^-*psb*A tends to be next to a G4-*psb*A or a G4-*psb*A pseudogene in *F. thermalis*, except when D1^INT^-*psb*A is located at the end of the contig. In contrast, G4-*psb*A was not found next to D1^INT^-*psb*A in the mesophilic *Fischerella* genomes (Appendix A). The genomic context for the other isoforms and sequence alignment files can be found in the Appendix A. These results suggest that some isoforms and gene copies of D1/*psb*A are selected by high-temperature conditions; thus, we then investigated whether the transcription of photosynthesis-associated genes was also affected by temperature.

### 2.2. Effect of Temperature on the Transcription of Photosynthesis-Related Genes in Fischerella thermalis

To determine the effect of temperature on photosynthetic transcriptional activity in the cyanobacterium *F. thermalis*, we recovered and analyzed the transcriptional levels of photosynthesis-related genes along the thermal gradient (66, 58, 48 °C) of Porcelana hot spring (Figure 2A) using metatranscriptomes already available in the literature [8]. The results show that most of the genes associated with photosynthesis, such as the subunits of PSII (e.g., *psb*A and *psb*D) and PSI (e.g., *psa*A and *psa*B), exhibited higher levels at 48 °C or 58 °C, while some genes also related to respiration, such as those encoding ATP synthase or electron carriers, had higher levels at 58 °C or 66 °C. In the case of reaction center genes, the transcript levels of *psb*A and *psb*D (encoding for D1 and D2, respectively) were likely to be lower at 58–66 °C (Figure 2A). However, owing to a lack of data that prevented more robust and specific analyses, we decided to evaluate the transcriptional levels of the D1^INT^-encoded *psb*A and *psb*D under controlled laboratory conditions using cultures of strain *F. thermalis* PCC 7521 grown at 45 °C (Section 4.2). The subsequent cultures were subjected to different temperature treatments from 40 °C to 65 °C lasting from 30 min to 6 h. The heat-shock responsive gene *gro*EL [51] showed increased transcript levels at higher temperatures (Appendix A), suggesting that high-temperature treatments were affecting the stability of cellular components and triggering a heat-shock response [52,53,54,55], which probably affected the photosynthetic apparatus [55,56]. In the case of the D1^INT^-*psb*A gene (Figure 2B), increasing transcript levels were observed from 40 °C to 55 °C, with a 15.28-fold increase after 1 h of treatment, whereas transcript levels decreased at 60 °C to 65 °C. This indicates that D1^INT^-*psb*A transcription probably peaks near the photosynthetic limit of *F. thermalis* (~58 °C), but is reduced above this temperature. In contrast to D1^INT^-*psb*A, the maximum *psb*D transcript levels were observed at 55 °C and 60 °C (Figure 2C), suggesting that D1 (specifically D1^INT^) and D2 respond differently to temperature. In addition, the increase in *psb*D transcripts was lower, showing less than a threefold change at the peak levels compared with that at 40 °C, and its transcripts were overall 10- to 60-fold lower than those of the D1^INT^-*psb*A gene. We also found that the transcript levels of genes associated with PSI (e.g., *psa*A, *psa*B and *psa*F; Figure 2A) were reduced under high temperatures, suggesting that cyclic electron flow in *F. thermalis* is also affected by temperature. Because carbon-based energy pathways, especially those related to electron transfer, are suggested as auxiliary pathways in cyanobacteria under stress [43] and because we found indications that respiration is transcriptionally induced by temperature (Figure 2A), we decided to evaluate alternative energy pathways for photosynthesis and the co-expression of energy metabolism genes in *F. thermalis*.

### 2.3. Alternative Energy Pathways and Co-Expression of Energy Metabolism Genes in Fischerella thermalis

To analyze whether alternative energy pathways for photosynthesis are involved in the high-temperature response of *F. thermalis*, and, because cyanobacteria are reported to store carbon and energy as carbohydrates [57], we analyzed in silico the transcriptional levels of genes related to glycolysis in the metatranscriptomes of the thermal gradient (48, 58, 66 °C) at Porcelana hot spring (Figure 3A). As a result, we found a heterogeneous transcriptional pattern, consistent with the fact that some of these genes are involved in both carbon fixation and carbon consumption processes. Key genes related to glycolysis, such as *pfk* and *pyk* (encoding glycolysis-associated enzymes 6-phosphofructokinase and pyruvate kinase, respectively), presented higher transcriptional levels at 58 °C or 66 °C in Porcelana hot spring (Figure 3A). Furthermore, analysis of these same genes in *F. thermalis* PCC 7521 cultures demonstrated higher *pfk* levels at 55 °C and 60 °C for most evaluated time points (Figure 3B), with a 7.12-fold peak at 60 °C after 1 h of treatment, which is similar to what was previously found for the *psb*D gene (Figure 2C). In the case of *pyk* (Figure 3C), transcriptional levels in the *F. thermalis* PCC 7521 cultures were more variable over the treatment time. Specifically, higher levels were observed at low temperatures during the 30 min treatment; the levels then increased at 50–60 °C after 1 h of treatment; and finally, no statistical differences were found after 3 or 6 h of treatment. As transcriptional levels of glycolysis-related genes are likely to increase at high temperatures and, as in cyanobacteria, pathways associated with carbon usage are connected to photosynthesis [43,58], we explored if this induction is part of a photosynthesis-related response in *F. thermalis*.

We also evaluated whether genes for energetic metabolism are co-expressed in *F. thermalis* within the context of the thermal gradient. To that end, we analyzed metatranscriptomic sequences associated with *F. thermalis* from 11 samples collected within the El Tatio geothermal field (Atacama, Chile) at hot springs representing temperatures between 45 °C and 57 °C (Appendix A). The abundance of *F. thermalis* in the El Tatio hot springs was more variable than that in the Porcelana hot spring (Patagonia, Chile), and the number of retrieved sequences varied among samples (1 × 10^5^ to 1 × 10^6^). For this reason, we did not analyze a direct association with temperature, but instead measured transcriptional correlation levels of the genes. Because we were only interested in metabolic genes, we restricted the analysis to those genes affiliated with enzyme-associated proteins in the BioCyc database [59]. As a result, we identified 710 genes that had an *r* value ≥ 0.85 with at least one other gene. Based on the correlation levels, we built a correlation network that exhibited a total of 3679 connections forming a total of 19 clusters with at least 5 genes (Figure 4A). Using the BioCyc platform [59], we analyzed which metabolic pathways were overrepresented in clusters 1, 6, 7, 8, and 12 (Appendix A), corresponding to the five clusters with higher number of nodes. We found that clusters 1, 7, and 8 were associated with the energy metabolism of *F. thermalis* (Figure 4B and Appendix A). In general, cluster 1 was associated with amino acid metabolism and glycolysis, suggesting an association with carbon metabolism, whereas clusters 7 and 8 were strongly associated electron transfer processes. Cluster 7 contained genes associated with the PSII reaction center and cytochrome b_6_f subunits, indicating a primary association with photosynthesis. Finally, cluster 8 harbored genes associated with NDH-1 and cytochrome C oxidase, as well as some structural PSII genes, indicating an association with both photosynthesis and respiration.

Additionally, to gain a better understanding of how the energy metabolism of *F. thermalis* is connected to the thermal gradient of the El Tatio hot springs, we reconstructed a metabolic network of *F. thermalis* based on the BioCyc database (Appendix A). Some steps of the TCA-cycle were curated manually based on the literature (See Section 4.8). Subsequently, we merged our correlation data with the energy metabolism network of *F. thermalis* (Figure 5). As expected, PSII was mostly associated with clusters 7 and 8, as well as with the cytochrome C oxidase and NDH-1 complexes. Conversely, glycolysis was mostly associated with cluster 12, which is related to sugar metabolism (Appendix A). These results show a strong association between photosynthesis and respiration, at both the transcriptional and functional levels in *F. thermalis*.

### 2.4. Effect of Temperature on the Photosynthetic Efficiency of Fischerella thermalis

Thus far, our results have shown that photosynthesis and carbon-related pathways are probably overexpressed near the reported photosynthetic limit of *F. thermalis*. However, it was necessary to understand how this is related to the photochemical activity of PSII. To that end, we evaluated the effect of temperature on PSII efficiency using cultures of the *F. thermalis* strain PCC 7521. Our preliminary assays showed that the Fm values obtained were lower than those obtained of Fp (data not shown), so we opted to measure the Kautsky effect under continuous light conditions. The resulting Fv values were calculated using Fp-F_0_ instead of Fm-F_0_ [60]. Quenching-associated measurements were discarded. As in previous *F. thermalis* PCC 7521 gene transcription experiments performed in this study, the cultures were subjected to different temperature treatments from 40 °C to 65 °C for 2 h, after which chlorophyll fluorescence was measured. To obtain reliable Fv/Fp measurements, *F. thermalis* PCC 7521 was kept under dark conditions and pretreated with the PSII inhibitor DCMU for 2 h before temperature treatment, as has been reported in studies with other cyanobacteria [61,62].

As a result, we found that the Fv/Fp of *F. thermalis* was significantly affected by temperature after 2 h of treatment (Figure 6 and Table 1 and Table 2). *Fischerella thermalis* was most efficient at 40 °C and 45 °C (Fv/Fp = 0.27). The Fv/Fp value then decreased with increasing temperature to 0.18 and 0.17 at 50 °C and 55 °C, respectively, and finally to 0.11 at 60 °C, while no fluorescence was detected at 65 °C. The loss of fluorescence is consistent with the transcriptional decrease in the *psb*A and *psb*D genes at 60–65 °C, but not with the levels observed at lower temperatures (Figure 2B,C). The highest Fv/Fp value of *F. thermalis* was 0.27, which was lower than the maximum photosynthetic yield reported for other cyanobacteria (usually about 0.6 [61]). As our results were based on Fp and not Fm, further studies based on Fm are needed to compare the photosynthetic efficiency of *F. thermalis* to other cyanobacteria. 

## 3. Discussion

### 3.1. Possible Loss of D1 Gene Copies and Retention of the Photoresistant D1^INT^ Isoform in Fischerella thermalis

Photosynthesis is the main energy metabolism mechanism of cyanobacteria. Our results indicate that photosynthesis in *F. thermalis* is negatively affected by temperature. We found that the copy number of the classical G4 D1 isoform of the PSII reaction center in *F. thermalis* was lower than other non-thermal cyanobacteria of the genus *Fischerella* (Figure 1B). This may suggest a process of genomic optimization, as genome reduction has previously been associated with adaptation to thermal environments in other bacteria [63]. Most genomes belonging to the genus *Fischerella* were highly fragmented (Appendix A), thus it is also possible than the lower G4 copy number in *F. thermalis* is because of assembly related issues, especially considering the high similarity of the *psb*A-related sequences. However, we found that some copies of G4-*psb*A found only in mesophilic *Fischerella* were located near transposase-coding genes (Figure 1D), suggesting that these potential transposon-associated G4-*psb*A copies were either lost during the *F. thermalis* speciation process or originated as part of a genome expansion exclusive to mesophilic *Fischerella*. Conversely, two of the three *psb*A genes encoding the G4 isoform in *F. thermalis* were located near other photosynthesis-related genes, such as phycobilisome-rod-encoding genes and the D1^INT^-*psb*A gene (Figure 1C). This suggests that high-temperatures conditions may have favored the clustering of photosynthesis-related genes rather than the transposon-associated G4-*psb*A copies. The synteny of the G4-*psb*A copies was consistent between the genomes of the mesophilic *Fischerella* and the same was observed for the *F. thermalis* genomes (Appendix A). Thus, this strongly suggests that the observed differences are due to differences between the genomes of *F. thermalis* and the mesophilic *Fischerella*. Still, the low availability of unfragmented *Fischerella* genomes prevented us from fully confirming loss of the G4-*psb*A copies. As such, further studies on the evolution of the genus *Fischerella* are now required to fully understand the genome optimization process of *F. thermalis*. Interestingly, D1^INT^ and D1^FR^ isoforms were previously recognized as part of the G4 group [38], but have been more recently suggested as new groups positioned between G3 and G4 [39]. Previous studies indicate that *F. thermalis* can use far red-light for photosynthesis by undergoing a complex acclimation process known as far-red light photoacclimation (FaRLiP), which leads to the formation of the red-shifted chlorophylls *d* and *f* and the incorporation of these pigments into newly synthesized photosystems [64,65,66]. Therefore, the presence of D1^FR^ (Figure 1), which is associated with the FaRLiP gene cluster [65], is expected in combination with G1-D1 encoding the chlorophyll *f* synthase [67] and the phycobilisome-linker protein ApcE2 [5,39,65]. Relative to the standard G4 isoform, D1^INT^ [39] possesses specific amino acid changes that interact with electron acceptors within the PSII reaction center [39,68,69,70]. Hence, the increased D1^INT^-*psb*A transcription could be associated with changes in the electron transfer activity, as we found that electron transfer chain activity and carbon metabolism also respond to high temperatures (Figure 2, Figure 3, Figure 4 and Figure 5). Although the physiological characteristics of D1^INT^ require further study, it is noteworthy that this potentially photoresistant D1 isoform is retained in the thermal cyanobacterium *F. thermalis.* As temperature can aggravate PSII photoinhibition in photosynthetic organisms [71], we hypothesize that retention of the D1^INT^ isoform may be associated with a strategy to overcome adverse thermal conditions by reducing the associated negative consequences rather than increasing PSII thermotolerance.

### 3.2. Photosynthesis-Related Genes Are Differentially Regulated by Temperature in Fischerella thermalis

The photosynthesis-related genes in *F. thermalis* seem to be largely down-regulated by temperature (Figure 2A), except for *psb*A and *psb*D, whose transcript levels increased with increasing temperature up to ~60 °C (Figure 2B,C). In this study, we found that Fv/Fp, which can be associated with the light used by PSII under non-saturation conditions, decreases with temperature in *F. thermalis* (Figure 6), being consistent with previously reported photosynthetic analyses for *F. thermalis* [72,73]. Although the measured Fv/Fm in cyanobacteria is usually lower than the effective Fv/Fm [62], our Fp-based results show a clear effect of high temperature over Fv/Fp. Given the fact that PSII is highly susceptible to high-temperature stress [14], our results suggest that PSII complexes in *F. thermalis* are being destabilized by high temperature. Both the Fv/Fp measurements and the overall transcriptional pattern observed for photosynthesis-related genes are consistent with a decreasing number of PSII complexes as temperature increases. However, the increased transcription of *psb*A and *psb*D is not consistent with a decrease in the photosynthetic efficiency. Furthermore, the transcriptional levels of *gro*EL suggest an active heat shock response at high temperatures (Appendix A), which probably affects the stability of cellular components and the photosynthetic process [52,53,54,55,56]. A possible explanation for this is that the transcription of *psb*A and *psb*D is related to specific D1 and D2 temperature-associated turnover requirements rather than the photosynthetic efficiency. It is known that D1 participates in photochemical reactions, thereby requiring higher turnover rates than D2 to maintain PSII [37]. Thus, the transcription of *psbA* and *psb*D could be increased to compensate for the temperature-induced destabilization of PSII, which would fulfill their turnover requirements and maintain PSII activity over a wide temperature range. At temperatures above 60 °C, thermal damage to PSII may be irreparable or the metabolic cost of maintaining functional photosynthesis may be higher than the energy produced, thus leading to loss of photosynthetic activity. Another possibility is that photosynthesis-related genes are post-transcriptionally regulated by non-coding RNA (ncRNA) [74]. Although it has been reported that photosynthesis-related genes in cyanobacteria are mostly transcriptionally regulated [75], the inconsistency uncovered in the present study between F_v_/F_m_ and *psb*A activity could also be explained by post-transcriptional regulation. Further experiments focusing on, for example, ncRNA and the stability of cellular components (such as mRNA and proteins) should be conducted to understand the role of post-transcriptional regulation on the high-temperature-response of *F. thermalis*.

### 3.3. Emerging Evidence of Mixotrophic Metabolism by Fischerella thermalis in Relation to Its Thermotolerance

Transcriptional data have revealed that genes related to glycolysis are induced by high-temperature stress in cyanobacteria, such as *Synechocystis* sp. PCC 6803, [14], while auxiliary energy pathways have been proposed to be relevant for regulating photosynthesis under stress conditions [43]. The mixotrophic behavior of *F. thermalis* has been previously reported [17,18], but how this relates to its thermotolerant capacities is unknown. In the present study, we found emerging evidence that the mixotrophic metabolism of *F. thermalis* could be relevant to its thermotolerance. First, we found that, although the transcriptional pattern of glycolysis is variable among genes, the *pfk* gene, which directs carbohydrates for glycolysis, was induced by high temperature (Figure 3B), suggesting that temperature is associated with organic carbon usage. Furthermore, we corroborated that respiration and photosynthesis are not only metabolically connected, but are also co-expressed processes (Figure 4 and Figure 5). As respiration plays a major role in thylakoid redox regulation and photorespiration, thus preventing photoinhibition [41,42], this co-expression suggests that it is likely related to the regulation and/or optimization of photosynthesis at high temperatures. However, the transcriptional levels of other genes, such as *pyk*, were not found to be related to temperature; thus, there is still a lack of evidence to support the hypothesis that sugars are being consumed by *F. thermalis* in response to high temperatures. Further studies are thus required to uncover whether the carbon metabolism-related response is related only to regulation and optimization of photosynthesis or whether there is a genuine mixotrophic response in *F. thermalis*.

Overall, our results are consistent with a compensatory photosynthetic response to high temperatures in *F. thermalis*. Increased transcription of reaction center genes in this organism would allow it to repair its temperature-damaged PSII reaction centers, while respiration prevents damage to the photosynthetic apparatus. Furthermore, even though more information is needed in this regard, the results of this work suggest that respiration and carbon metabolism may help obtain energy from organic carbon sources in the environment. Finally, the selection of potentially photoresistant PSII reaction centers in *F. thermalis* could help mitigate the negative consequences of living at high temperatures. This thermal adaptation strategy by *F. thermalis* conceptually relies on transcriptional and metabolic regulation rather than on major changes in protein sequences (although it is compatible with them), which could be especially relevant for highly conserved complexes like PSII. This study provides novel information on the response of *F. thermalis* to the high temperatures of hot springs and proposes a molecular model of photosynthesis in high-temperature ecological niches. While there are several aspects of this model that remain to be demonstrated (including the biochemical properties of D1^INT^ and the post-transcriptional regulation of D1 isoforms), we expect that future studies will help us to better understand the photosynthetic response of *F. thermalis* to high temperatures.

## 4. Materials and Methods

### 4.1. Phylogenetic Reconstruction and Analysis of the D1 Isoforms of Fischerella thermalis

*Fischerella* genomes were retrieved from the NCBI RefSeq and GenBank assembly databases using a taxonomic search at the Hapalosiphonaceae family level (NCBI taxid: 1892263; accessed December 2021). Next, the genomic taxonomy was determined using GTDB-tk v0.3.2 software with database version R202 [76]; finally, 48 genomes [44,45] distributed in 7 *Fischerella* species were selected for further analyses. The cyanobacteria *Gloeobacter violaceus* PCC 7421 (NCBI genome ID: GCF_000011385.1), *Nostoc* sp. PCC 7120 (NCBI genome ID: GCF_000009705.1 and GTDB R202 species *Trichormus* sp000009705), and *Chlorogloeopsis fritschii* PCC 9212 (NCBI genome ID: GCF_000317265.1) were used as outgroups, because these species have been studied in relation to D1 [38,39,46]. An annotated *psb*A G1 isoform from *F. thermalis* PCC 7521 (WP_009455234.1) was searched in outgroup genomes using DIAMOND v2.0.5 (BLASTp -k 1 -e 0.0001 –query-cover 70) [77]. Subsequently, these sequences were used to perform another DIAMOND search (BLASTp -k 1000 -e 0.0001 --query-cover 70) against the 48 *Fischerella* and the 3 outgroup genomes. The retrieved *psb*A sequences were aligned using MUSCLE v6.0.98 software [78]. Maximum-likelihood trees were generated with the IQtree v.1.5.5 software [79]. The best substitution model was selected using the TESTNEW option (ModelFinder) [80] and tree reconstruction was performed using ultrafastbootstrap support of 10,000 replicates [81]. Isoform group identities G0-to-G4 were obtained from the literature [38]. D1^INT^ and D1^FR^ were identified by sequence similarity to their respective consensus sequences [39]. Protein sequence presence and counts for the resulting *psb*A were mapped into a phylogenomic tree obtained by multiple sequence alignment of 122 bacterial markers from the GTDB-tk v0.3.2 [76] analysis and maximum likelihood tree reconstruction with IQtree v.1.5.5 software [79,81], as described above, using *Nostoc* sp. PCC 7120 as an outgroup species. Resulting trees were adjusted with the iTOL web server [82]. *psb*A genes are summarized in Appendix A. D1 isoform sequences were grouped according to the corresponding GTDB R202 genome species assignation. The genomic context for D1^INT^ and G4 was analyzed by selecting five upstream and five downstream genes for each D1^INT^-*psb*A and G4-*psb*A gene copy from the genomes of *F. thermalis* NIES-3754 and *Fischerella* sp. NIES-4106. The genome fragments were plotted using the MG2C platform [83].

### 4.2. Fischerella thermalis PCC 7521 Culture Conditions and Temperature Treatments

Axenic cultures of *F. thermalis* PCC 7521 were grown in BG11_0_ media at 45 °C and 15.5 µmol m^−2^ s^−1^ (6500K LED light, 12/12 h light/dark cycle) until they reached ~24 mg chlorophyll/L. Next, 15 mL of culture was placed in individual 25 mL flasks 3 h after turning on the lights to ensure active photosynthesis. Cultures were then immediately subjected to temperature treatments at 40 °C, 45 °C, 50 °C, 55 °C, 60 °C, or 65 °C for 30 min, 1 h, 3 h, or 6 h. Three biological replicates per time/temperature were used. The cultures were subjected to ultracentrifugation at 8000× *g* for 15 min, after which the supernatant was discarded, and then the pellets were stored in liquid N_2_ until analysis.

### 4.3. RNA Extraction

Eleven samples from El Tatio geothermal field (45 °C to 57 °C) were collected (Appendix A) and stored in RNALater™ (Thermo Fisher, Waltham, MA, USA). RNA from *F. thermalis* PCC 7521 cultures and the El Tatio environmental samples was extracted with 1 mL of TRIzol™ (Thermo Fisher, Waltham, MA, USA) and 250 µL of chloroform, which was added for phase-separation, followed by centrifugation at 12,000× *g* for 15 min. RNALater™ from the El Tatio samples was diluted by washing with sterile double distilled water prior to TRIzol™ addition. After phase-separation of the TRIzol™-embedded samples, the aqueous phase was transferred to a 2 mL tube and amended with 1 volume of molecular-grade ethanol. The mixtures were subjected to the RNA Clean and Concentrator kit (Zymo Research, Irvine, CA, USA), according to the manufacturer’s protocol. To prevent genomic DNA contamination, the RNA samples were incubated for 40 min with DNAse from the “Turbo DNAse kit” (Invitrogen, Waltham, MA, USA), as per the manufacturer’s protocol. 

### 4.4. cDNA Synthesis

cDNA was synthetized from 1 µg of DNA-free RNA of *F. thermalis* PCC 7521 using the ImProm-II Reverse Transcription System kit (Promega, Madison, WI, USA) according to the manufacturer’s protocol. To improve the specific cDNA synthesis, we used a mix of reverse primers targeting the genes of interest [84]. Primers were designed using Primer3 [85] based on the *F. thermalis* NIES-3754 genome. Primers are listed in Appendix A.

### 4.5. qPCR Analysis

qPCRs were set up using the qPCR Master Mix (2X) kit (KAPA Biosystems, Wilmington, MA, USA), according to the manufacturer’s protocol. Reference genes were obtained from the literature [86,87]. qPCRs were performed on the LightCycler 480 real-time PCR platform (Roche, Basel, Switzerland). The resulting data were analyzed and graphed in Graphpad Prism 9 (https://www.graphpad.com/; GraphPad Software, USA).

### 4.6. Metatranscriptomic Analysis of Porcelana Hot Spring and El Tatio Environmental Samples

RNA-seq libraries of eleven metatranscriptomes from El Tatio geothermal field samples representing temperatures of 45 °C to 57 °C (Appendix A) were prepared and sequenced at the Roy J. Carver Biotechnology Center (University of Illinois at Urbana-Champaign, Champaign, IL, USA). The KAPA HyperPrep kit (Roche, Basel, Switzerland) was used for library preparation and sequencing was performed on the Illumina NovaSeq 6000 on flowcell S1 (2 × 150 bp). Metatranscriptomes representing temperatures of 48 °C, 58 °C, and 66 °C from Porcelana hot spring were retrieved from the NCBI SRA database (SRA ID: SRP104009 [19]). Sequences belonging to *F. thermalis* were retrieved by alignment with previously assembled bacterial metagenome-assembled genomes (MAGs) from hot spring samples [5,8] using DIAMOND (e-value ≤ 1 × 10^−7^) [77]. Reads assigned to non-*Fischerella* species were discarded. Remaining sequences were double-checked by aligning them to the gene set of the recovered *F. thermalis* MAGs, as well as the *F*. *thermalis* NIES 3754 and *F*. *thermalis* PCC 7521 proteins using DIAMOND (e-value ≤ 1 × 10^−7^). Proteins were functionally assigned using eggNOG mapper [88]. Gene counts were summarized according to their respective KEGG ortholog (KO) ID. Gene counts were normalized by quantiles using the command normalize.quantiles() from the R package preprocessCore with default parameters [89].

### 4.7. Co-Expression Network Analysis and Pathway Overrepresentation

*Fischerella thermalis* gene count correlation was analyzed using the R package WGCNA [90]. Metabolic genes were identified using the BioCyc database [59], with *F. thermalis* PCC 7521 as a reference. Gene pairs with an *r* ≥ 0.85 were selected as co-expressed genes. The co-expression network was reconstructed in Cytoscape [91] and submitted for cluster analysis with the Cytoscape package ClusterMaker2 [92]. Isolated gene clusters comprising less than five genes were not considered as clustered. Pathway overrepresentation was then analyzed for the main clusters (see results). Overrepresented pathways for the selected clusters were depicted as a word cloud plot using the R packages ggplot2 [93] and ggwordcloud [94].

### 4.8. Metabolic Network of *Fischerella thermalis* Reconstruction

Metabolic pathways of *F. thermalis* PCC 7521 were obtained from the BioCyc database [59]. As cyanobacteria use succinic semialdehyde as an alternative to succinyl-CoA [95], SSADH and 2-OGDC coding genes were identified in *F. thermalis* PCC 7521 by sequence similarity to *Synechococcus* sp. PCC 7002 orthologs (SynPCC7002_A2771 and SynPCC7002_A2772) using BLASTp (-k 1000 -e 1e-7 --query-cover 70). Metabolic pathways were parsed as a table using R. The metabolic network was visualized using Cytoscape [91]. To improve network visualization, ATP, GTP, ADP, GDP, and H_2_O nodes were removed from the network owing to their extensive connectivity.

### 4.9. Chlorophyll Fluorescence Measurement of Temperature-Treated *Fischerella thermalis* PCC 7521 Cultures under Continuous Light Conditions

Axenic cultures (15 mL) from *F. thermalis* PCC 7521 grown under the same conditions as those used for the temperature treatments were placed in 25 mL flasks under dark conditions overnight. Cultures were pretreated for 2 h with either dimethyl sulfoxide (DSMO; Sigma-Aldrich, St. Louis, MO, USA)) as the control or 3-(3,4-dichlorophenyl)-1,1-dimethylurea (DCMU; Thermo Fisher; 1 × 10^−5^ M (10 mM stock solution)). The cultures were then subjected to a 2 h temperature treatment at 40 °C, 45 °C, 50 °C, 55 °C, 60 °C, or 65 °C. Cultures were under dark conditions until fluorometry. Chlorophyll fluorometry was measured using the FluorCam HFC 1000-H fluorometer (Photon Systems Instruments, Drásov, Czech Republic) adjusted to 30% sensitivity with the FluorCam 7 software (Photon Systems Instruments). The default F_0_ Kautsky protocol of the FluorCam 7 software (available in the Appendix A) was used to measure Kautsky kinetics under continuous light conditions. The F_0_ from *F. thermalis* PCC 7521 cultures acclimated in darkness was measured for 5 s, then the Fp was measured by irradiating the cultures with actinic light (light irradiance 1000 µmol m^−2^ s^−1^) for 30 s. Then, the Fv/Fp value was calculated as Fp-F_0_/Fp [60]. The results were analyzed by two-way ANOVA in R [96] and graphed with Graphpad Prism 9 (https://www.graphpad.com/; GraphPad Software).

## Figures and Tables

**Figure 1 molecules-27-08515-f001:**
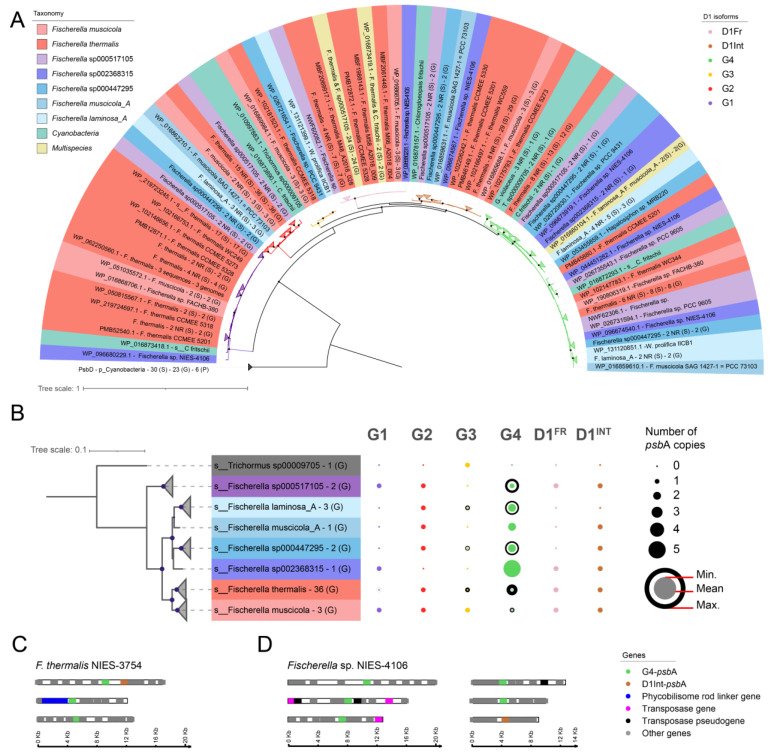
D1 isoforms within the genus *Fischerella.* (**A**) Phylogenetic reconstruction of 126 *psb*A sequences from 48 *Fischerella* genomes. The outgroups comprised D1 sequences from *Gloeobacter violaceus* PCC 7421 (NCBI genome ID: GCF_000011385.1), *Nostoc* sp. PCC 7120 (NCBI genome ID: GCF_000009705.1 and GTDB R202 species *Trichormus* sp000009705), and *Chlorogloeopsis fritschii* PCC 9212 (NCBI genome ID: GCF_000317265.1), as well 30 *psb*D sequences. Branches are colored according to the D1 isoform, while labels are colored according to GTDB R202 *Fischerella* species, sequences from outgroup genomes, or identical sequences present in more than one species. For collapsed clades, the labels show the number of sequences, number of non-redundant sequences (i.e., identical NCBI protein ID), and number of genomes. Bootstrap support values over 90% are shown as black dots in the tree nodes. (**B**) Phylogenomic reconstruction of 48 *Fischerella* genomes and one outgroup (*Nostoc* sp. PCC 7120). Tree leaves are collapsed at species level and colored according to (**A**). Bootstrap support values over 95% are shown as black dots in the tree nodes. Theisoform count was performed according to the distribution of *psb*A sequences in (**A**), where the outer, middle, and inner circles represent the maximum, average, and minimum number of *psb*A sequences copies, respectively. (**C**,**D**) To analyze the genomic context of the G4 (green) and D1^INT^ (brown) isoforms of the *psb*A gene from *F. thermalis* NIES-3754 (**C**) and *Fischerella sp.* NIES-4106 (**D**), a region consisting of 5 genes upstream of the *psb*A gene, the respective *psb*A, and five genes downstream of the *psb*A gene was selected as the genome fragment. Genes are represented by boxed colored as per the legend.

**Figure 2 molecules-27-08515-f002:**
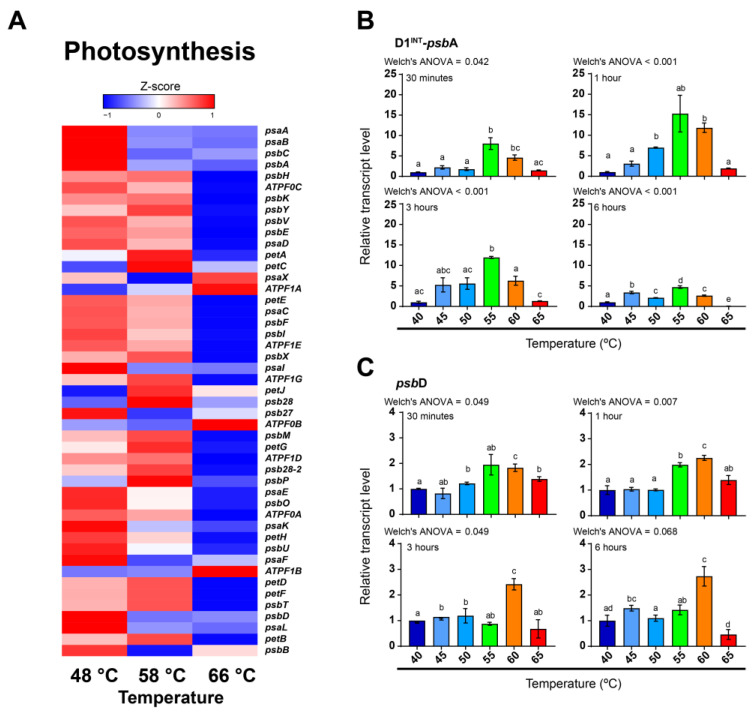
Transcriptional activity of photosynthesis-related genes in *F. thermalis* at different temperatures. (**A**) Based on Porcelana metatranscriptomes [19], the transcriptional levels of photosynthesis-related genes were analyzed at 48 °C, 58 °C, and 66 °C. Genes with multiple copies or isoforms and their respective transcript levels were grouped as a single gene. The Z-score color-scale indicates transcriptional levels from low (blue) to high (red). (**B**,**C**) The transcriptional levels of *psb*A (**B**) and *psb*D (**C**) in *F. thermalis* cultures treated with temperatures from 40 °C to 65 °C (*n* = 3) were analyzed by RT-qPCR. Relative transcript levels were adjusted to 1 using 40 °C as a reference for each analyzed time. Error bars denote the standard error of the mean (SEM). Barplots are colored for aesthetic purposes. Welch’s ANOVA was used for statistical analysis. Different lowercase characters above each bar denotes significant differences between temperature treatments.

**Figure 3 molecules-27-08515-f003:**
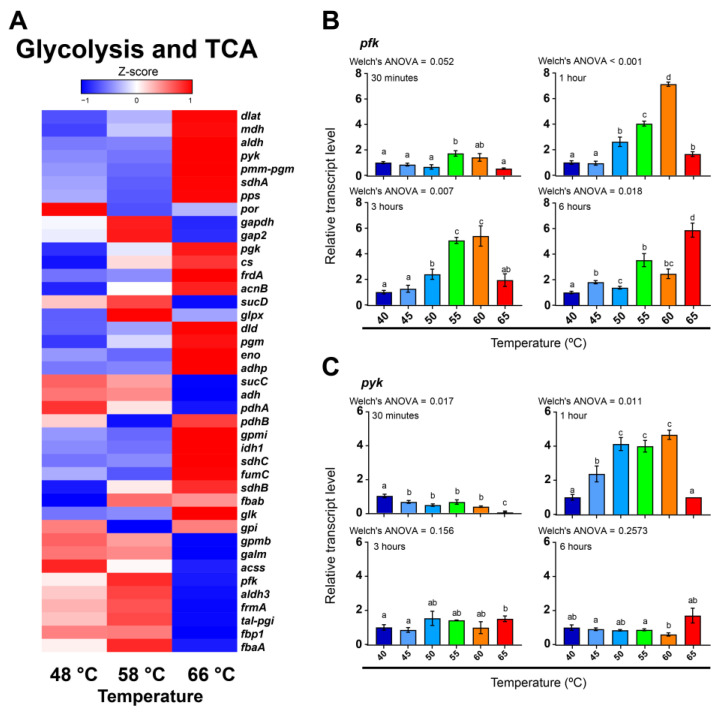
Transcriptional activity of glycolysis-related genes in *F. thermalis* at different temperatures. (**A**) Based on Porcelana metatranscriptomes [19], the transcriptional levels of photosynthesis-related genes were analyzed at 48 °C, 58 °C, and 66 °C. Genes with multiple copies or isoforms and their respective transcript levels were grouped as a single gene. The Z-score color-scale indicates transcriptional levels from low (blue) to high (red). (**B**,**C**) The transcriptional levels of *pfk* (**B**) and *pyk* (**C**) in *F. thermalis* cultures treated with temperatures from 40 °C to 65 °C (*n* = 3) were analyzed by RT-qPCR. Relative transcript levels were adjusted to 1 using 40 °C as reference for each analyzed time. Error bars denote the SEM. Barplots are colored for aesthetic purposes. Welch’s ANOVA was used for statistical analysis [19]. Different lowercase characters above each bar denotes significant differences between temperature treatments.

**Figure 4 molecules-27-08515-f004:**
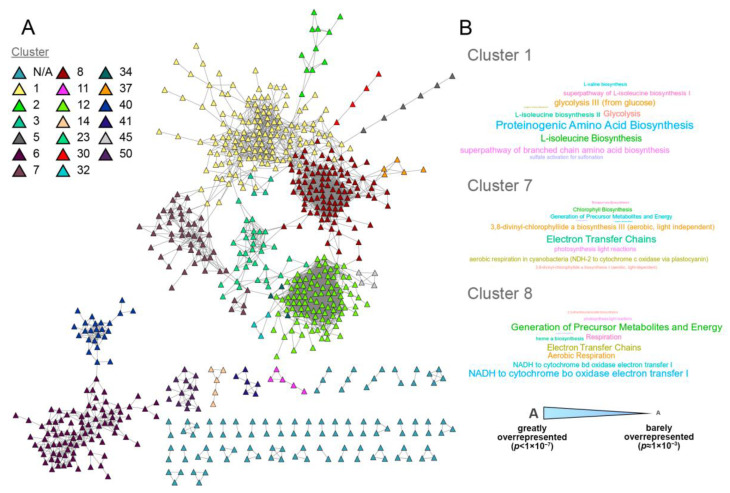
Gene co-expression network and overrepresented metabolic pathways. (**A**) The gene co-expression network was reconstructed from metabolic genes that were transcriptionally correlated (*r* ≥ 0.85) with at least one other metabolic gene. The resulting network comprised 710 genes and 3697 links, with 19 identified clusters denoted by different colors. Node labels were omitted to prevent text overlapping. Network Cytoscape file can be found in the Appendix A. (**B**) Overrepresented metabolic pathways for clusters 1, 7, and 8 were analyzed and are represented as word clouds. Font size denotes the degree of overrepresentation, from minimally overrepresented (small font) to greatly overrepresented (big font).

**Figure 5 molecules-27-08515-f005:**
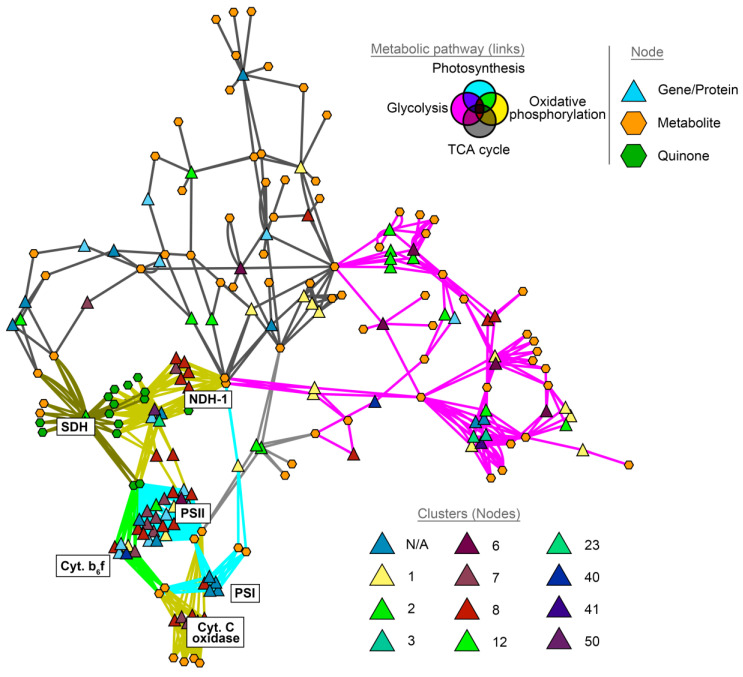
Co-expression of energy metabolism genes in *F. thermalis*. Gene co-expression clustering (Figure 4) was integrated with the energy metabolism network of *F. thermalis* (Appendix A). The node color denotes the corresponding gene co-expression cluster. The link color denotes the corresponding metabolic pathway, with intermedial colors indicating an association with more than one pathway. Node labels were omitted to prevent text overlapping. Respiration and photosynthesis complexes are indicated with white labels. Network Cytoscape file can be found in the Appendix A.

**Figure 6 molecules-27-08515-f006:**
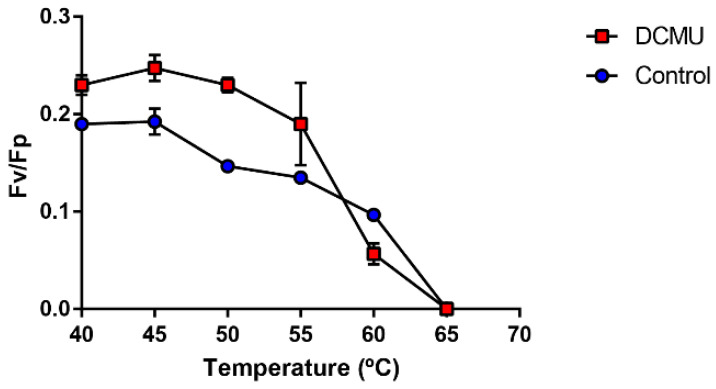
Maximum photosynthetic yield (Fv/Fp) of *F. thermalis* at different temperatures. Fv/Fp was determined by measuring chlorophyll fluorescence in *F. thermals* cultures after 2 h of treatment at temperatures from 40 °C to 65 °C. A 2-h preincubation with DCMU (red boxes) or DSMO-control (blue circles) was employed for better Fp measurements. Error bars denote SEM. *n* = 3.

**Table 1 molecules-27-08515-t001:** Summary of two-way ANOVA for Fv/Fp measurements of temperature-treated *F. thermalis* cultures.

Factor	F-Value	*p*-Value
Temperature	14.0456	<0.0001
DCMU	1.5343	0.2188
Temperature/DCMU	2.6381	0.0286

**Table 2 molecules-27-08515-t002:** Summary of Tukey’s test for Fv/Fp measurements of temperature-treated *F. thermalis* cultures.

Temperature (°C)	Diff	Lower	Upper	Adj. *p*-Value
45–40	0.0390115	−0.009507	0.0875305	0.1740
50–40	−0.018833	−0.070941	0.0332745	0.8500
55–40	−0.026833	−0.078941	0.0252745	0.6040
60–40	−0.089833	−0.141941	−0.037725	<0.0001
50–45	−0.057845	−0.102187	−0.013503	0.0043
55–45	−0.065845	−0.110187	−0.021503	0.0008
60–45	−0.128845	−0.173187	−0.084503	<0.0001
55–50	−0.008	−0.056242	0.0402425	0.9900
60–50	−0.071	−0.119242	−0.022758	0.0009
60–55	−0.063	−0.111242	−0.014758	0.0043

## Data Availability

Metatranscriptomic sequences associated with *F. thermalis* from El Tatio samples can be found under NCBI BioProject accession number PRJNA885895.

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
