# Peer review of "Compensatory Transcriptional Response of Fischerella thermalis to Thermal Damage of the Photosynthetic Electron Transfer Chain"

_molecules, 2022, doi:10.3390/molecules27238515_

Round 1

Reviewer 1 Report

The authors present an interesting and well-executed study concerning the regulation of PS-related genes in function of elevated abiotic stress, in this case temperature, in the thermotolerant cyanobacterium F. thermalis. Based on (meta)transcriptional analysis, they suggest this response occurs at the transcriptional level and propose linkage to gene transcriptional activity in specific metabolic pathways.

A number of caveats need attention:
- the phylogenetic analysis of psbA encoded D1 sequences has been a hot topic for many years, and this across all photosynthetic organisms; from this much has been learned about the functional and evolutionary importance of specific aa residues and aa substitutions in D1. Although the authors show the phylogenetic distribution of six D1 isoforms within the Fischerella genus and make the interesting observation that D1-G4 is underrepresented in F. thermalis, they do not provide any insights into the exact molecular diversity at the aa level i.e. have D1 residues been identified that are highly specific to F. thermalis and may be correlated to thermotolerance? The underlying multiple alignments for Fig.1 are not made available to the reader, neither as Jalview-generated figures (e.g. shareable via https://figshare.com/) nor in MSA format (PHYLIP, MSF,..).
- much of the manuscript centers around transcriptional regulation yet the authors refrain from any discussion in regard to known regulatory factors and UR elements of studied genes, such as, e.g. for psbA, sigmafactors, RpoD3, CmpR, 6S rRNA, sRNA, asRNA, -35 UPE, etc. Did the authors look at UR's of psbA (or other genes) in an attempt to identify common UR transcriptional elements?
- the issue of variable mRNA stability under abiotic stress (temperature) has not been discussed
- besides transcription, expression levels may also depend on post-transcriptional regulation (e.g., inhibition of translation); the authors do not discuss this
- for metatranscriptional analysis, RNAseq was used; however, the detection, identification and associative comparison of ncRNA's (are some more dominant in F. thermalis?) is not being discussed
- the manuscript suffers from vagueness in what is or can be concluded

Minor remarks:
- L38: starts with 2x 'The'
- L112: perhaps best to list here which 3 other genera?
- L133: be careful with the term 'repressed' (it implies negative control, which may not be the case?)
- L178: Porcelana should not be in Italic
- L183: should be 'analysis'
- Fig. 2: psbA is listed twice, perhaps add isoform notation?
- L269: what is meant with "loss" - perhaps a typo? (should be "lost"?)
- L357-360: what is the rationale of choosing these 3 organisms?
- L374 (and previous item): please note that PCC 7120 is highly resistant to gamma radiation, another form of abiotic stress
- L381: which light source was used, with which spectrum and photon flux density?
- L381: in photobiology, and specifically when in the PAR of the e.m. spectrum, the use of microeinstein per second and square meter (microE.m-2 .s-1) is generally advocated; this may depend on Journal Guidelines though
- L41209: should be 2x instead of 2X?
- L389-426: for ALL equipment and ALL consumables, make sure to adhere to (company, city, country), or for USA based use (company, city, state); only first time out, thereafter just company suffices;
- L443: which BLAST program was used (standalone, web, ..?), where, how, parameters, ..reference?
- L437: it is customary to put R packages and other script names in courier format
- L 450-51: make sure to explain all abbreviations (DCMU, DSMO)
- L455: should be Fo Kausky? (o subscript?)
- there doesn't seem to be a M&M section for chlorophyll fluorescence measurements (procedures, equipment, ..)

2.13.0.0 2.13.0.0

Reviewer 2 Report

This manuscript reports genetic activity related to the energy metabolism of the cyanobacterium Fischerella thermalisboth in situ and in cultures at high temperatures (40°C to 65°C). The molecular and bioinformatics approaches, in addition to photophysiological analyses, were employed for this study. It was found that photosynthesis of the cyanobacterium decays at high temperatures. The increased transcriptional activity of genes encoding photosystem II reaction center proteins compensate thermal damage up to 60°C. The cyanobacterium F. thermalis tends to lose copies of the standard G4 D1 isoform while maintaining the recently described D1INT isoform; this suggests a preference for photoresistant isoforms in response to the thermal gradient. The work showed that F. thermalis uses several compensatory adaptations to cope with the harsh conditions of hot springs.

The comment is listed below.

1.     “The The global optimal temperature for oxygenic photosynthesis is estimated be-38 tween ~10°C and ~30°C [1–4].”; typo error at “The The global optimal”.

Author Response

We greatly appreciate the contribution of the three reviewers to the process of improving the manuscript. All comments have been considered and we now provide a new version of the manuscript indicating here our responses and changes that have been made in each section.

Thank you for your really nice opinion, it really cheered us up.

The typo at the start of the introduction was corrected in line 37.

Cheers.

Reviewer 3 Report

The authors tried to identify the diversity of sequence and copy number of the psbA gene isoforms in the genomes of cyanobacterium Fischerella thermalis. They also tried to identify the expression of the psbA isoformes in thermal damage-dependent manner. Their working hypothesis sounds interesting, but their technical approaches were not suitable to solve their questions.

The most genomic data used in this phylogenetic analysis were derived from public assembly data generated by Miller et al. (2020) Current Biology 30, 344–350 (doi.org/10.1016/j.cub.2019.11.056) which was also discussing about adaptive evolution of the genus Fischerella. However, the above reference was not cited in this manuscript. Please cite the reference and discuss the genomic plasticity of the genus Fischerella.

Most of public assembly data of the genus Fischerella thermalis were low-quality draft genomes. Some of them includes more than 1000 contigs despite the bacterial small genome. Assembly strategies of these draft genome sequences were very simple and their protocols were not considering the paralogues genes with highly identical sequences such as the psbA isoforms. Therefore, it is difficult to say about the loss of psbA isoforms in specific strains without additional biological experiments and/or reanalysis of the de novo assembly with careful approach to detect the isoforms of the psbA gene and other highly-homologous multi-copy genes. I worry that the authors are blindly trusting the sequences in public database too much. In this point of view, Figure 1 and its discussion includes some severe problems.

In such case, the analysis should be based on the complete genome sequence of some other Fischerella species.

Line 67: I’ve never heard the re-definition of the species name of Thermosynechococcus elongatus BP-1. And reference (40) was not the officially-recognized journal for the definition of species name.  

Line 98 and Figure 1: The results in Figure 1 should be compared with the phylogenetic tree of 16S rRNA.

Line 101: The word, “0.6 copy/genome” was not sound meaningfully. It should be changed to “30/50 strains have single copy of the G4 isoform, but others did not possess the isoform gene”or another sentence.

Line 128: The authors described that the psbA gene was exhibited at 48 degree C or 58 degree C. As they know, paralogues of the psbA genes have highly identical in their nucleotide sequences and usually some of them are constitutively expressed and others are expressed in stress-dependent manner. In their transcriptome analysis, how they detected the expression levels of the individual transcripts of the psbA paralogues? Especially, in the metatranscriptome analysis, how did the authors destinguish the individual transcripts of the psbA paralogues derived from closely related species? The protocols stated in line 413-437 is not enough to solve the above-mentioned problems.

Line 176 Figure 2: Please enlarge the text font in this figure. Functions of the typical genes in Fig. 2A should be indicated next to the gene ID.

Line 269-291. The authors discussed about the loss of the D1 protein or functions of D1 isoforms. However,

Additional physiological approaches such as proteins level experiment must be necessary to discuss about the participation of the various isoforms of D1 proteins in the thermal damage of PS machineries.

Round 2

Reviewer 1 Report

the manuscript has been significantly improved and I am happy to note greater availability of data; nonetheless, some very minor textual issues remain, some a bit more critical than others..

L52: should be "..has been found at temperatures as high as 66 °C.." (detected may assume molecular detection, if this indeed is the intention keep 'detected' instead of 'found')
L138: owing to textual changes, the sentence "This suggests that temperature.." does not connect very well with previous sentences?
L144: should be "PCC 7120" (space after PCC)
L217: there is something amiss with this sentence: "..of the glycolysis..", perhaps should be 'of glycolysis genes', or 'of glycolysis-related genes'?
L244: "in the El Tatio field" (add article)
L387: should be "it is reported"
L391: something is missing after "cellular"? (mRNA?)
L430: better may be "high-temperature"?
L432: either "a better understanding of the" or just "understand the"
L442: should be "PCC 7120" (space after PCC)
L445: either "this species has" or "these species have"
L483: no need to repeat, just use "(Thermo Fisher)"
L505: you may opt to add "(https://www.graphpad.com/") instead
L567: should be "the Fischerella genus" (add article)
general: why is El Tatio sometimes regular and sometimes italic?

Fig. 3: I missed this in the first round of review, but SEM has nowhere explained as 'Standard Error of the Mean', so it would be best to introduce the term here

Fig. S2: it occurs to me that no legends are given for the supplementary materials; particularly for Fig. S2 this is a shortcoming as I cannot readily relate colors or notations with observations/statements in the manuscript (or the authors' rebuttal); in fact, the legends of Figs. 2 and 3 do not contain information on the meaning of used colors and notations; I suggest extending those legends accordingly, and at least add the same information at L572, i.e., when referring to Fig. S2. The simplest way to do this, is to improve the legend to Fig.1, then accordingly refer to this legend in the legend of Fig. 2, and once again on L572 - make sure all notations are explained.

2.13.0.0

Reviewer 3 Report

 The revised manuscript was improved largely. I think additional few revisions must be necessary for publcation.

>The most genomic data used in this phylogenetic analysis were derived from public assembly data generated by Miller et al. (2020) Current Biology 30, 344–350 (doi.org/10.1016/j.cub.2019.11.056) which was also discussing about adaptive evolution of the genus Fischerella. However, the above reference was not cited in this manuscript. Please cite the reference and discuss the genomic plasticity of the genus Fischerella.

>R: Our mistake, we apologize. The authors appreciate the reviewer's comment, and the missing reference was added. It is worth mentioning that although the genomes used in this study come from 14 different references, 30 of the 49 genomes used in this study come from only two articles: Sano et al. (2018) (https://doi.org/10.1038/s41559-017-0435-9) and Miller et al. (2020) (https://doi.org/10.1016/j.cub.2019.11.056). Regarding genomic plasticity, we found a high association between two isoform pairs in Fischerella thermalis, which are not associated in other Fischerella species and therefore this is now discussed in lines 109-127 and 307-322.

[reviewer’s comment]

Line 327-334. It seems an over-speculation to connect the difference of partial synteny of the photosynthetic genes with the thermo-tolerance without other experimental evidences.

It might help the readers if the references papers of Fischellera with complete genome sequence are added to the section.

>Most of public assembly data of the genus Fischerella thermalis were low-quality draft genomes. Some of them includes more than 1000 contigs despite the bacterial small genome. Assembly strategies of these draft genome sequences were very simple, and their protocols were not considering the paralogues genes with highly identical sequences such as the psbA isoforms. Therefore, it is difficult to say about the loss of psbA isoforms in specific strains without additional biological experiments and/or reanalysis of the de novo assembly with careful approach to detect the isoforms of the psbA gene and other highly-homologous multi-copy genes. I worry that the authors are blindly trusting the sequences in public database too much. In this point of view, Figure 1 and its discussion includes some severe problems.In such case, the analysis should be based on the complete genome sequence of some other Fischerella species.

>R: We appreciate this comment. The genomes used in this study have over 96 % completeness according to CheckM v1.0.18 software (Parks et al., 2015; https://doi.org/10.1101/gr.186072.114), and 5 genomes with lower completeness were left out of the study.

[reviewer’s comment]

Completeness score of CheckM cannot solve this problem completely. Since short-read sequencing data is difficult to escape from the problem of de novo assembly of multicopy genes, the authors should be checking carefully when they construct the story using public data of draft genome. It is still difficult to declare “the loss” of the psbA isoforms in specific strains by the approach in this version of manuscript. The psbA gene and it’s isoforms have traditionally attracted attention by many researchers. Therefore, the authors should be careful with that description. Please add information like that “most of the 48 genomes are fragmented “at line 102-103. Please  weaken the claim in the paragraph title  of “3.1. Fischerella thermalis loss D1 gene copies…” to “3.1. Fischerella thermalis possibly loss D1 gene copies…”

Line 128: The authors described that the psbA gene was exhibited at 48 degree C or 58 degree C. As they know, paralogues of the psbA genes have highly identical in their nucleotide sequences and usually some of them are constitutively expressed and others are expressed in stress-dependent manner. In their transcriptome analysis, how they detected the expression levels of the individual transcripts of the psbA paralogues? Especially, in the metatranscriptome analysis, how did the authors destinguish the individual transcripts of the psbA paralogues derived from closely related species? The protocols stated in line 413-437 is not enough to solve the above-mentioned problems.

R: Given the high similarity between the D1 isoforms and their respective psbA gene, we decided to group the genes according to their function and their counts were pooled in sections 2.2 and 2.3. We have only kept the qPCR result for the psbA gene as specific to D1INT, since in this case specific primers were used for this gene, whose sequence is available in the table S4

[Reviewer’s comment]

Line 522. Please add description of the parameter settings in the R package preprocessCore.
